Root bacterial endophytes alter plant phenotype, but not physiology

Henning Jeremiah A. 1 jhennin2@vols.utk.edu
Weston David J. 2
Pelletier Dale A. 2
http://orcid.org/0000-0002-7804-5044 Timm Collin M. 3
Jawdy Sara S. 2
Classen Aimée T. 1 4
1 Department of Ecology & Evolutionary Biology, University of Tennessee–Knoxville , Knoxville, Tennessee , United States
2 Biosciences Division, Oak Ridge National Laboratory , Oak Ridge, TN , United States
3 Joint Institute for Biological Sciences, University of Tennessee , Oak Ridge, TN , United States
4 Center for Macroecology, Evolution, and Climate, The Natural History Museum of Denmark, University of Copenhagen , Copenhagen , Denmark
Day David
Electronic publication date: 2016 Nov 1
Publication date: 2016
Volume: 4
Electronic Location ID: e2606
Received 2016 Jun 27; Accepted 2016 Sep 24
Copyright: © 2016 Henning et al.
Copyright year: 2016
Copyright holder: Henning et al.
License: This is an open access article distributed under the terms of the Creative Commons Attribution License, which permits unrestricted use, distribution, reproduction and adaptation in any medium and for any purpose provided that it is properly attributed. For attribution, the original author(s), title, publication source (PeerJ) and either DOI or URL of the article must be cited.
License URL: https://creativecommons.org/licenses/by/4.0/

Keywords: Bacterial endophytes, Burkholderia, Plant functional traits, Populus trichocarpa, Pseudomonas fluorescens, Trait plasticity, Plant morphology

Funding: U.S. DOE Office of Biological and Environmental Research, Genomic Science Program UT-Battelle, LLC, for the US Department of Energy DEAC05-00OR22725 U.S. Department of Energy, Office of Science, Office of Biological and Environmental Research, Terrestrial Ecosystem Sciences Program DE-SC0010562 Funding from Plant-Microbe Interfaces Scientific Focus Area project at Oak Ridge National Laboratory was provided by the U.S. DOE Office of Biological and Environmental Research, Genomic Science Program. Oak Ridge National Laboratory is managed by UT-Battelle, LLC, for the US Department of Energy under contract no. DEAC05-00OR22725. JH was supported, in part, by the U.S. Department of Energy, Office of Science, Office of Biological and Environmental Research, Terrestrial Ecosystem Sciences Program under Award Number DE-SC0010562. The funders had no role in study design, data collection and analysis, decision to publish, or preparation of the manuscript.

==============================
Plant traits, such as root and leaf area, influence how plants interact with their environment and the diverse microbiota living within plants can influence plant morphology and physiology. Here, we explored how three bacterial strains isolated from the Populus root microbiome, influenced plant phenotype. We chose three bacterial strains that differed in predicted metabolic capabilities, plant hormone production and metabolism, and secondary metabolite synthesis. We inoculated each bacterial strain on a single genotype of Populus trichocarpa and measured the response of plant growth related traits (root:shoot, biomass production, root and leaf growth rates) and physiological traits (chlorophyll content, net photosynthesis, net photosynthesis at saturating light–Asat, and saturating CO2–Amax). Overall, we found that bacterial root endophyte infection increased root growth rate up to 184% and leaf growth rate up to 137% relative to non-inoculated control plants, evidence that plants respond to bacteria by modifying morphology. However, endophyte inoculation had no influence on total plant biomass and photosynthetic traits (net photosynthesis, chlorophyll content). In sum, bacterial inoculation did not significantly increase plant carbon fixation and biomass, but their presence altered where and how carbon was being allocated in the plant host.

Introduction

A recent review exploring microbiome-mediated plant traits found that plant-associated microbes can modify 14 out of 30 commonly measured functional traits (Cornelissen et al., 2003; Friesen et al., 2011). For example, inoculation with common root-colonizing bacterial strains influenced root and leaf architectural traits, such as specific leaf area and specific root length, as well as plant physiological traits such as carbon fixation and chlorophyll content (Harris, Pacovsky & Paul, 1985; Ma et al., 2003; Friesen, 2013). Further, inoculation by different members of the plant microbiome may differentially alter plant phenotype (Zamioudis et al., 2013; Timm et al., 2016). The presence of unique bacterial strains in legume genotypes explained more variation in shoot biomass, root biomass, and plant height than plant genotype did (Tan & Tan, 1986). Inoculation of common endophytes can also inhibit primary root elongation and promote lateral root formation and root hair production (Zamioudis et al., 2013; Weston et al., 2012). Recent breakthroughs in the multitude of the −omics fields have allowed for unprecedented mechanistic investigations of microbe-induced changes in host function (Verhagen et al., 2004; Walker et al., 2011; Weston et al., 2012; Vandenkoornhuyse et al., 2015; Timm et al., 2015; Timm et al., 2016) and have been the subject of multiple recent reviews (Friesen et al., 2011; Friesen, 2013; Vandenkoornhuyse et al., 2015; Hacquard & Schadt, 2015; Lebeis, 2015; and many others). This work demonstrated that plant growth promoting bacteria elicit numerous changes in host gene expression through multiple and simultaneous hormonal and immune response pathways (Verhagen et al., 2004; Walker et al., 2011; Weston et al., 2012; Drogue et al., 2014; Timm et al., 2016). However, these studies fall short in explaining how changes in gene expression influence the overall plant phenotype or plant function. Thus, understanding the response of plant traits and overall plant phenotype to microbial strains remains a research gap.

Here, we inoculated three endophytic bacterial strains (Pseudomonas fluorescens GM41, Pseudomonas fluorescens GM30, and Burkholderia sp. BT03), originally isolated from wild Populus, on a single genotype of Populus trichocarpa and measured plant phenotypic response to bacterial inoculation. We measured a suite of traits commonly measured in the functional trait ecology literature to explore how phenotype is influenced by bacterial strains within the pre-existing functional trait framework. Plant functional trait ecology has largely ignored microbiome contribution to plant phenotype. Bacterial strains belonging to the Pseudomonas fluorescens group are common plant growth promoting bacteria that are abundant in the Populus microbiome (see Gottel et al., 2011). Pseudomonas fluorescens accounted for approximately 34% of the sequences found in the Populus endosphere, but only 2–3% of the sequences in the rhizosphere and soil samples originating from the same roots (Gottel et al., 2011). Pseudomonas strains can alter plant host function by modifying plant growth (Kloepper et al., 1980; Lugtenberg & Kamilova, 2009; Timm et al., 2015), nutrient allocation (Bisht et al., 2009), hormone signaling (Stearns et al., 2012), up-regulating/down-regulating of gene expression pathways (Timm et al., 2016), and immune function (Verhagen et al., 2004; Weston et al., 2012). Additionally, the Pseudomonas fluorescens clade has a large amount of functional diversity (Jun et al., 2016), thus selecting two Pseudomonas strains allows us to explore how plant traits and overall phenotype respond to closely related bacterial strain genomes. To contrast with these two strains, we selected a distantly related, but enriched in Populus endosphere (Gottel et al., 2011), bacterial strain from the genus Burkholderia.

We predicted that aboveground and belowground traits of Populus trichocarpa would respond to Burkholderia and Pseudomonas strains and inoculation of different bacterial strains would result in different plant phenotypes. Further, we predicted that the two Pseudomonas strains would produce a plant phenotype that was more similar to one another than to Burkholderia because of phylogenetic relatedness, i.e. more shared functionality. To test our predictions, we first conducted a genomic comparison using clusters of orthologous groups (COG) database to predict the functional differences among strains. Next, we inoculated each bacterial strain on Populus trichocarpa and measured a suite of physiological and architectural plant traits including the root:shoot, biomass production, root and leaf growth rates, chlorophyll content, net photosynthesis, and net photosynthesis at saturating light–Asat, and saturating CO2–Amax. We chose to measure overall trait response to bacterial endophytes without measuring the pathways involved because we were interested in understanding down-stream consequences of bacterial inoculation on overall plant phenotype.

Materials and Methods

Populus trichocarpa genotype “93–968” (Labbé et al., 2014) was propagated in tissue culture following standard procedures (see Kang et al., 2009). Briefly, in vitro cultures were established from actively growing shoot tips collected from greenhouse-grown Populus plants. We sterilized shoot tips by soaking fresh cut tips in a 1% Tween 20 solution for 5 min, 70% Ethanol solution for 1 min, a 0.525% sodium hypochlorite solution for 15 min and then rinsed them three times in sterile H2O for 5 min. Shoot tips were trimmed to 2 cm in length and transferred to a magenta box (Sigma-Aldrich, St. Louis, MO, USA) containing 80 ml of tissue media (1× Murashige & Skoog (MS) basal medium (Murashige & Skoog, 1962) supplemented with MS vitamins (Caisson Labs, North Logan, UT, USA), 0.05% 2-(N-morpholino) ethanesulfonic acid (MES hydrate) (Sigma-Aldrich, St. Louis, MO, USA), 3% sucrose, 0.1% PPM™ (plant protective mixture) (Plant Cell Technology, Washington, DC, USA), 0.5% activated charcoal (Sigma-Aldrich, St. Louis, MO, USA), and 0.15% Gelzan (Plantmedia, bioWORLD, Dublin, OH, USA). Plants were sub-cultured until it was determined, using microscopy and colony formation units with R2A medium, that the plants were axenic.

Plant cultures were rooted in a growth room at 25 °C under a 16 h photoperiod. After root establishment, plants that were similar in size and developmental stage were selected for experimentation. Plants were weighed and scanned to account for initial plant size differences among treatments. To ensure sterility during scanning, plants were placed between two (21.59 × 27.94 cm) sheets of cellulose acetate that were sprayed with 100% ethanol. Scans were performed with a portable scanner (VuPoint Solutions Inc., City of Industry, CA, USA) at 600 × 600 dpi. Scanned images were analyzed in WinRhizo (Regent Instruments, Quebec City, Canada) to determine initial root surface area, root length, stem length, and leaf surface area. After scanning, plants were transferred into experimental microcosms.

Experimental design

We constructed closed microcosms by interlocking two sterile Magenta boxes (Sigma-Aldrich, St. Louis, MO, USA) with a coupler (Sigma-Aldrich, St. Louis, MO, USA). We added 150 ml calcined clay (Pro’s choice Sports Field Products, Chicago, IL, USA) and 70 ml of 1× Hoagland’s nutrient solution (Sigma-Aldrich, St. Louis, MO, USA) to each microcosm. We drilled two 7 mm holes on adjacent sides of the upper magenta box and covered the holes with adhesive microfiltration discs (Tissue Quick Plant Laboratories, Hampshire, United Kingdom) to allow air to flow into and out of the microcosms and to prevent outside microbial contamination. Prior to microbial addition, we double sterilized each closed microcosm by autoclaving on a 60 m dry cycle on consecutive days. Pseudomonas fluorescens strains (GM30 and GM41) and Burkholderia sp. (BT03), hereafter termed Pseudomonas GM30, Pseudomonas GM41, and Burkholderia BT03 were isolated from Populus deltoides endospheres from east Tennessee and western North Carolina, USA (originally described in Brown et al., 2012). For full isolate descriptions, see Brown et al. (2012), Weston et al. (2012), Utturkar et al. (2014), Timm et al. (2015) and Timm et al. (2016). We selected these three strains because previous work (Pseudomonas GM30–Weston et al., 2012; Labbé et al., 2014; Pseudomonas GM41–Labbé et al., 2014; Timm et al., 2016; Burkholderia Bt03–Timm et al., 2016) had given us indication that strains were able to influence traits in Arabidopsis thaliana (Weston et al., 2012), were able to manipulate plant gene expression and hormonal signaling in P. deltoides (Timm et al., 2015; Timm et al., 2016), and were able to influence host interactions with mycorrhizal symbionts (Labbé et al., 2014). Although strains were isolated from P. deltoides, strains from Pseudomonas and Burkholderia readily colonize natural P. trichocarpa tissues (Moore et al., 2006; Xin et al., 2009; Knoth et al., 2014; Khan et al., 2014; Doty et al., 2016). We grew bacterial strains in isolation and at a constant temperature, 25 °C, in 5 ml of R2A medium. After growing overnight they were pelleted and re-suspended in sterile water to an OD600 of 0.01 (∼1.0 × E7 cells ml−1).

We inoculated each microcosm by adding 10 ml of the bacterial strain (107 cells ml−1) to the calcined clay substrate and stirring for 30 s to distribute the bacteria. After inoculation, we planted the Populus clones within each microcosm. Each Populus was grown in an individual microcosm in combination with one of the bacterial strains. Thus, the experiment had four treatment combinations–Pseudomonas GM30 inoculation, Pseudomonas GM41 inoculation, Burkholderia BT03 inoculation, and a bacteria-free control. In total, there were 32 microcosms with four treatments (n = 8). The experiment was divided into three different establishment dates in 2014 (1 March, three replicated blocks; 25 March, two replicated blocks; and 2 April, three replicated blocks) because microbiome-free plant tissues were difficult to propagate. Plant-bacteria combinations were grown in the microcosms for five weeks with a 16 h photoperiod, at 21 °C and 80% relative humidity.

After 35 days of growth, plants were removed from microcosms, submerged in sterilized deionized H2O to remove clay from the root system, weighed, and scanned. Scans were analyzed with WinRhizo to determine final root surface area, total root length, stem length, and leaf surface area. For each plant, the final measurement of root surface area, total root length, stem length, and leaf surface area was subtracted from the initial measurement and divided by the experiment duration to determine tissue growth rates (cm d−1 or cm2 d−1). Additionally, each plant was dried for 48 h at 70 °C and weighed to measure leaf, shoot (leaf + stem) and root and total dry mass. Specific leaf area and the specific root length of each individual were calculated by dividing leaf area by leaf dry mass or by dividing root length by root dry mass, respectively.

To measure host physiological response to different bacterial strains, leaf gas-exchange was measured and used to estimate leaf photosynthesis on our first replicate block (March 1, n = 3). For each plant, gas exchange of the largest leaf of the plant was measured (Li-Cor model 6400, Li-Cor Biosciences, Lincoln, NE, USA) immediately prior to our experimental harvest. The maximum rate of photosynthesis in saturating light under ambient CO2 (Asat), the maximum rate of photosynthesis in saturating light and saturating CO2 (Amax), and the quantum yield of CO2 fixation (Φ) were all measured. Finally, average leaf chlorophyll content was measured on three fully opened leaves (Konica Minolta Chlorophyll Meter SPAD-S02, Ramsey, NJ, USA).

Comparative genomics of microbes

Genomes of Pseudomonas GM30 and GM41 and Burkholderia BT03 were sequenced at Oak Ridge National Laboratory and genes were identified using Prodigal (Brown et al., 2012; Utturkar et al., 2014) and are available at NCBI (GM41: AKJN00000000.2; GM30: NZ_AKJP00000000.2; BT03: NZ_AKKD00000000.2). Genome annotation, genomes statistics, and annotation comparisons were performed using IMG tools (https://img.jgi.doe.gov/). Genome statistics and COG functional predictions were extracted from Integrated Microbial Genomes (https://img.jgi.doe.gov/) and then they were compared manually for differential inclusion of predicted functions.

Bacterial colonization

To test for endophytic colonization of Pseudomonas GM41, Pseudomonas GM30, and Burkholderia BT03, we planted cuttings of P. trichocarpa into a magenta box using similar methodology and treatments described above (n = 3). After two weeks of growth, all the plant roots, stems, and 1–2 mature leaves were surface sterilized by dipping them in a ∼10% bleach solution, followed by 70% ethanol, and then rinsing in water three times. We recorded wet weight of plant tissues and then separately macerated each plant tissue compartment in a sterile mortar and pestle in 1 ml sterile 1× PBS. We transferred macerated plant tissues to a 24-well plate where we serial diluted each sample by 10% with 1× PBS at 1×, 0.1×, 0.01× of original sample concentration. Each sample was streaked onto R2A media plates and allowed to grow for 48 h at 20 °C. After 48 h, colony formation was counted. We calculated CFU mg−1 of plant tissue by multiplying colony number per plate by 10(dilution factor + 1) and then dividing that number by the dry tissue mass (mg1).

Statistical analyses

We tested all data for normality using the normalTest function in the fBasics package (version 3011.87; R Metrics Core Team, 2014) for R version 3.0.2 (R Development Core Team, 2013) and RStudio version 0.98.495 (RStudio Team, 2013). If data were not normally distributed, we performed log transformations or square-root transformations to satisfy the normality assumptions of analysis of variance (ANOVA).

To explore plant trait response (root dry mass, leaf dry mass, shoot dry mass, total dry mass, root: shoot, root growth rates, leaf growth rates, change in leaf number, specific root length, specific leaf area) to bacterial strains, we used linear mixed-effect models using the lme4 package in R (Bates et al., 2015). Bacterial strain was a fixed effect in the model and experimental block (three establishment dates) was a random factor. For plant dry mass measures, we incorporated initial measurements of root surface area in the root dry mass model and initial leaf surface area in the aboveground dry mass model as covariates. To test for significance of bacterial strain (fixed effects) and covariate (initial growth measure) we performed a likelihood ratio test to compare models with and without fixed effects and covariates. If including fixed factors (bacterial strain) was significant an improvement to model fit (p < 0.05 in likelihood ratio test), we calculated least square means and confidence intervals using the difflsmeans function to calculate differences among strains using the lmerTest package version 2.0-3 (Kuznetsova, Brockhoff & Christensen, 2014). We measured host response to bacterial inoculation by calculating the percent change in trait values ((mean trait value for Populus inoculated with bacterial strain − mean non-inoculated trait value) × 100).

To test physiological responses (carboxylase activity, Amax, Asat) of plant hosts to bacterial inoculation, we used one-way ANOVA using the Anova function in the CAR package, (version 2.0-22, Fox & Weisberg, 2011) because we collected physiology data on only a single sampling date (n = 3). All raw data and R code is available in Tables S2 and S3, respectively.

Results

Bacterial strains differ in genomic content

We compared the genomes of Burkholderia BT03 and Pseudomonas GM30 and GM41 based on predicted enzyme functions using the COG database (Table 1). Overall, our genome comparison demonstrated that the bacterial strains differed in genome size and functional gene content. Burkholderia BT03 had a relatively large genome (10.9 Mb) compared to Pseudomonas GM30 (6.1 Mb) and Pseudomonas GM41 (6.6 Mb) (Table 1). We found all three bacterial strains shared functions that were likely critical for establishment and survival in the plant microbiome including the production of the plant hormone auxin, pili, flagella, chemotaxis, increased signal transduction, and secretion systems. However, we found many functional differences among our strains. The genome of Burkholderia encoded multiple pathways predicted to be involved in the metabolism of the plant hormones, salicylate and ethylene (Table 1). Relative to the Pseudomonas genomes, the Burkholderia genome encoded for numerous secondary metabolite biosynthesis pathways and more carbohydrate and lipid transporters, suggesting increased metabolic capabilities within Burkholderia (Table 1).

Table 1 Predicted plant-interaction pathways in bacterial strains Burkholderia sp. BT03, Pseudomonas fluorescens GM30, and Pseudomonas fluorescens GM41.

Genome size, relevant pathways, and COG category statistics were identified using IMG tools. Where applicable, gene loci indicating predicted functions in genomes (individual genes or pathways) were included.

	Burkholderia BT03	Pseudomonas GM30	Pseudomonas GM41	
Genome size (Mb)	10.9	6.1	6.6	
ACC deaminase	PMI06_0002752	PMI25_02765	PMI27_01478	
Salicylate metabolism	PMI06_001931	NA	PMI27_05197	
Auxin biosynthesis	PMI06_005275	PMI25_03791	PMI27_00952	
Pili, fimbriae	PMI06_00372-3373	PMI25_00378-0372	NA	
Flagella	PMI06_009483-9498	PMI25_03624-3649	PMI27_02843-2866	
Chemotaxis	PMI06_009463-9475	PMI25_05665-5658	PMI27_05395-5382	
Type 2 secretion system	PMI06_001352-1341	PMI25_00837-00844	NA	
Type 3 secretion system	PMI06_000607-0617	NA	NA	
Type 4 secretion system	PMI06_009642-9622	NA	NA	
Type 6 secretion system	PMI06_001813-1833	PMI25_012011220	PMI27_02378-2397	
Carbohydrate metabolism (# of genes)	582	222	291	
Secondary metabolite metabolism (# of genes)	337	113	148	
Note:

NA = not applicable.

Even through Pseudomonas GM30 and Pseudomonas GM41 were classified as the same 16S OTU, their genome size differed as did the predicted functional capabilities of the two strains. The genome of Pseudomonas GM41 encoded for phosphorus solubilization and nitrate reduction, which were lacking in the Pseudomonas GM30 genome. Additionally, Pseudomonas GM41 contained more secondary metabolite biosynthesis elements compared to Pseudomonas GM30. We also found that the genome of Pseudomonas GM41 contained more genes coding for carbohydrate metabolism, lipid metabolism, and amino acid transport and metabolism, energy production and conversion, suggesting that Pseudomonas GM41 may contain more metabolic breadth than Pseudomonas GM30 (Table 1). Taken together, our results demonstrated that these three bacterial strains differ in genome size and their functional gene content.

Bacterial colonization of Populus root tissue

All three of the bacterial strains colonized Populus hosts. Colony-forming units were enriched in all three bacterial strains relative to the control in the 0.1× and 0.01× dilutions (0.1× dilution F = 18.77, p < 0.0001; 0.01× dilution F = 13.78, p < 0.0001, Table 2), although CFU number was variable across dilutions, tissue types, and bacterial strain. However, we found no difference in CFUs among non-inoculated control and Pseudomonas GM30, GM41, and Burkholderia BT03 inoculated host plants at the 1× dilution (F = 1.24, p = 0.319, Table 2). Across nearly all tissue types, we found that Pseudomonas GM30, Pseudomonas GM41, and Burkholderia BT03 inoculated plants had 10–10,000× more CFUs than did non-inoculated control plants (Table 2). All three bacterial strains colonized leaf and stem tissues, but the highest CFUs across bacterial treatments were consistently observed in roots (Table 2). Inoculated host plants contained 0–28,809,015 CFU mg−1 in roots, 0–1,166,273 CFU mg−1 in stems, and 0–73,537 CFU mg−1 in leaves compared to 0–400 CFU mg−1 in root tissues, 0 CFU mg−1 in stem tissue, 0–1,000 CFU mg−1 in leaf tissue compared to non-inoculated control plants (Table 2).

Table 2 Colony forming units found in leaf, root, and stem tissue of Populus trichocarpa genotypes inoculated with Pseudomonas GM30, Pseudomonas GM41, or Burkholderia BT03 across three different dilution factors: 1×, 0.1×, 0.01× concentrations of the original sample.

Pseudomonas GM41 and Burkholderia BT03 data were first published in Timm et al. (2016).

Treatment	Tissue	Dilution	mean CFU	St dev		Sum Sq	Df	F	p	
Control	Leaf	1.0E + 01	1,080.5	1,871.5	Bact.	1.3E + 12	3	1.24	0.319	
GM30	Leaf	1.0E + 01	19,574.7	30,672.7	Tissue	1.4E + 12	2	1.92	0.1699	
GM41	Leaf	1.0E + 01	1,141.3	1,809.3	B × T	1.6E + 12	6	0.74	0.6264	
BT03	Leaf	1.0E + 01	41,175.9	45,063.1	Resid.	8.1E + 12	24			
Control	Root	1.0E + 01	110.2	131.5						
GM30	Root	1.0E + 01	170,447.1	212,977.7						
GM41	Root	1.0E + 01	2,438.9	1,563.8						
BT03	Root	1.0E + 01	309,628.0	106,958.6						
Control	Stem	1.0E + 01	0.0	0.0						
GM30	Stem	1.0E + 01	1,166,273.0	1,872,593.0						
GM41	Stem	1.0E + 01	1,510.2	2,135.8						
BT03	Stem	1.0E + 01	654,513.2	688,365.7						
Control	Leaf	1.0E − 01	1,044.4	1,809.0	Bact.	1.2E + 13	3	18.77	> 0.001	
GM30	Leaf	1.0E − 01	16,643.8	28,827.9	Tissue	3.8E + 12	2	9.21	0.001	
GM41	Leaf	1.0E − 01	566.2	980.7	B × T	1.1E + 13	6	8.91	> 0.001	
BT03	Leaf	1.0E − 01	60,745.9	54,910.0	Resid.	5.0E + 12	24			
Control	Root	1.0E − 01	402.7	377.6						
GM30	Root	1.0E − 01	120,591.5	111,174.4						
GM41	Root	1.0E − 01	2,851.9	3,319.7						
BT03	Root	1.0E − 01	3,096,279.7	1,069,585.6						
Control	Stem	1.0E − 01	0.0	0.0						
GM30	Stem	1.0E − 01	289,189.7	330,089.7						
GM41	Stem	1.0E − 01	0.0	0.0						
BT03	Stem	1.0E − 01	904,314.7	1,099,508.6						
Control	Leaf	1.0E − 02	0.0	0.0	Bact.	6.9E + 14	3	13.78	> 0.001	
GM30	Leaf	1.0E − 02	0.0	0.0	Tissue	4.0E + 14	2	11.79	> 0.001	
GM41	Leaf	1.0E − 02	0.0	0.0	B × T	1.2E + 15	6	11.47	> 0.001	
BT03	Leaf	1.0E − 02	73,537.1	80,004.4	Resid.	4.1E + 14	24			
Control	Root	1.0E − 02	0.0	0.0						
GM30	Root	1.0E − 02	368,195.0	510,398.0						
GM41	Root	1.0E − 02	20,595.2	35,671.9						
BT03	Root	1.0E − 02	28,809,015.5	14,126,689.6						
Control	Stem	1.0E − 02	0.0	0.0						
GM30	Stem	1.0E − 02	227,127.9	252,544.5						
GM41	Stem	1.0E − 02	0.0	0.0						
BT03	Stem	1.0E − 02	1,805,855.2	1,567,125.7						

Plant structure is modified by bacterial inoculation

Overall, we found that plant trait response to bacterial endophytes was strain specific. Specifically, mean root growth rate increased 184% with Pseudomonas GM30 colonization (t = 3.84, p = 0.001), however root growth rates were unaffected by Pseudomonas GM41 (t = 1.61, p = 0.12), and Burkholderia BT03 (t = 1.18, p = 0.25) inoculation (Fig. 1; Table S1). Similarly, mean leaf growth rate increased 114 and 138% with Pseudomonas GM30 (t = 2.27, p = 0.03) and Pseudomonas GM41 (t = 2.86, p = 0.01) inoculation, but leaf growth rate was unaffected by Burkholderia inoculation (t = 1.02, p = 0.32) (Fig. 1; Table S1). Inoculation by Pseudomonas GM30 increased leaf number by 36% (t = 3.34, p = 0.003) but leaf number was unaffected by Pseudomonas GM41 (t = 0.93, p = 0.36) and Burkholderia BT03 (t = 1.418, p = 0.17) inoculation (Fig. 1). We observed no differences in stem elongation with bacterial inoculation (chisq = 0.06, p = 0.97, Table S1).

Figure 1 Structural traits of Populus trichocarpa that were not inoculated with bacteria (no microbe control) (n = 8), were inoculated with Pseudomonas GM30 (n = 7), Pseudomonas GM41 (n = 8), or Burkholderia BT03 (n = 7).

(A) Change in leaf number from the first to last day of the experiment. Negative values indicate that leaves senesced during the experiment. (B) Leaf surface area growth rates (cm2 d−1). (C) Stem growth rate (cm1 d−1). (D) Root surface area growth rates (cm2 d−1). Letters represent significant differences of post-hoc least squares means among bacterial treatments. Boxplots display median, first and third quartiles, and vertical lines represent 1.5× inner quartile range of our dataset. The dots represent raw data values.

Interestingly, we observed no differences in total plant dry mass (chisq = 3.27, p = 0.195, Fig. 2), root dry mass (chisq = 0.00, p = 1.00, Fig. 2), root:shoot ratio (chisq = 0.00, p = 1.00, Table S1) or plant height (chisq = 1.99, p = 0.158, Table S1) with bacterial inoculation. However, Pseudomonas GM30 inoculation increased leaf dry biomass by 86% (t = 2.43, p = 0.02) relative to control plants, however leaf biomass was unaffected by Pseudomonas GM41 (t = 0.97, p = 0.33) and Burkholderia BT03 (t = 1.70, p = 0.10) (Fig. 2; Table S1). We observed no differences in specific leaf area with bacterial inoculation (chisq = 2.60, p = 0.46, Table S1). Thus, inoculation of Pseudomonas GM30 increased leaf surface area (t = 2.27, p = 0.03) and aboveground dry mass (t = 2.43, p = 0.02), without changing leaf area:mass ratios. We found no significant differences in root length:dry mass (specific root length, chisq = 1.06, p = 0.79) with bacterial inoculation (Table S1). Our results indicate that bacterial strains modify plant resource allocation but not total dry mass production.

Figure 2 Biomass allocation of Populus trichocarpa that were not inoculated with bacteria (no microbe control) (n = 8), were inoculated with Pseudomonas GM30 (n = 7), Pseudomonas GM41 (n = 8), or Burkholderia BT03 (n = 7).

(A) Total dry mass (mg). (B) Leaf biomass (mg), (C) Root biomass (mg). Letters represent significant differences of post-hoc least squares means among bacterial treatments. Boxplots display median, first and third quartiles, and vertical lines represent 1.5× inner quartile range of our dataset. The dots represent raw data values.

Plant physiology is not affected by bacterial inoculation

Bacterial inoculation had no measureable effects on any physiological trait we measured: chlorophyll content (SPAD) (chisq = 2.15, p = 0.54), quantum yield of photosynthesis (ϕ) (F = 1.01, p = 0.43), net photosynthesis at saturating light conditions (Asat) (F = 0.76, p = 0.55) or maximum net photosynthesis at saturating light and [CO2] (Amax) (F = 1.98, p = 0.19) (Fig. 3). In agreement with the total dry mass data, we did not observe significant changes in the measured photosynthetic parameters. Thus, changes in plant structure were not linked with increased photosynthetic capacity, efficiency, or carbon assimilation rates.

Figure 3 Physiology traits of Populus trichocarpa that were not inoculated with bacteria (no microbe control) (n = 8), were inoculated with Pseudomonas GM30 (n = 7), Pseudomonas GM41 (n = 8), or Burkholderia BT03 (n = 7).

(A) Plant chlorophyll content (SPAD), (B) ΦCO2 (expressed as the slope of carboxylase activity across different light levels), and (C) carboxylase activity under maximum light level and CO2 concentration (Amax). Letters represent significant differences of post-hoc least squares means among bacterial treatments. Boxplots display median, first and third quartiles, and vertical lines represent 1.5× inner quartile range of our dataset. The dots represent raw data values.

Discussion

The plant root microbiome can have a strong influence on plant production and phenotype (Friesen, 2013; Vandenkoornhuyse et al., 2015); yet, less is known about how plant trait expression, production, and physiology are influenced by individual endophytic strains. We explored how plant morphological traits, productivity, and cellular physiology in Populus trichocarpa responded to inoculation with three bacterial strains, two closely related Pseudomonas fluorescens strains (GM30 & GM41) and a more distantly related Burkholderia strain (BT03). We selected bacterial strains that were predicted to differ in metabolic capabilities, plant hormone production and metabolism, and secondary metabolite synthesis in an effort to understand how plant phenotype is influenced by inoculation with different strains of common endophytic bacteria (Table 1, Timm et al., 2015; Timm et al., 2016). Our comparative genomic analysis revealed that while all three strains share many common endophytic functions like plant hormone signal disruption, production of plant hormone auxin, pili, flagella, and chemotaxis, strains potentially differed in their ability to perform these functions. Overall, we found that Burkholderia and Pseudomonas genomes differed in the carbon substrates they were predicted to degrade, plant hormone production and metabolism, and secondary metabolite synthesis, which led us to predict that plant response to bacterial inoculation would lead to different phenotypes between treatments. All three strains could colonize Populus roots, leaves, and stems; however, CFU number was highest within root tissues in all three strains (Table 2).

Overall, we found root endophyte inoculation altered plant resource allocation patterns without influencing total plant biomass accumulation (Fig. 1). Additionally, we found that plant trait response and overall phenotype differed across bacterial strains in ways that would not have been predicted from our genome analysis. Specifically, Burkholderia BT03 was predicted to produce auxin and to metabolize salicylate and ethylene, three plant hormones crucial to plant growth and development (see Yang & Hoffman, 1984; Wasternack & Parthier, 1997; Chen et al., 2009; Dempsey et al., 2011). Additionally, we found the Burkholderia genome encoded for multiple transposase elements that degrade poplar-produced aromatics and metabolites (Timm et al., 2015; Timm et al., 2016). Despite the predicted ability of Burkholderia to manipulate multiple plant hormonal and signaling pathways, we observed no measurable changes in any traits when Populus was inoculated with Burkholderia (Figs. 1–3). This was especially surprising since we consistently measured the highest CFU abundance within Burkholderia inoculated individuals (Table 2).

In spite of close genetic relatedness and classification under the same 16S OTU profile, our Pseudomonas strains differed in key functional capabilities. Specifically, Pseudomonas GM41 encoded for phosphate solubilization and denitrification ability, suggesting these two strains may differentially influence host nutrition, although this remains untested. Our genome analysis revealed that both strains were capable of producing the plant hormone auxin, however another study found that Pseudomonas GM41 produced two times more auxin than Pseudomonas GM30 (Timm et al., 2015). Auxin synthesis by endophytic bacteria can increase root branching and lateral root formation and decrease overall plant height, leaf number, chlorophyll content and photosynthetic efficiency (Romano, Cooper & Klee, 1993; Fu & Harberd, 2003; Weston et al., 2012). Thus, we predicted that Pseudomonas GM41 would have a strong influence on plant root traits, however we observed no measurable effects of Pseudomonas GM41 inoculation on root growth rate or morphology (Fig. 1; Table S1). Belowground, Pseudomonas GM30 inoculation increased root surface area growth rate by 184% (Fig. 1) without increasing root biomass (Fig. 2), suggesting Pseudomonas GM30 inoculation may change root morphology, leading to longer, thinner, highly-branched roots with similar biomass, as we predicted. Similar patterns have been observed when Pseudomonas GM30 is inoculated on both Arabidopsis (Weston et al., 2012) and Populus deltoides (Timm et al., 2015; Timm et al., 2016). Additionally, inoculation of Pseudomonas GM30 increased leaf surface area growth rate by 114% (Fig. 1), leaf number by 36% (Fig. 1), and aboveground biomass by 86% (Fig. 2) but did not influence specific leaf area (Table S1), whereas closely-related Pseudomonas GM41 increased leaf surface area growth rate by 138% (Fig. 1) but did not change leaf number (Fig. 1) or aboveground biomass (Fig. 2). Unlike Burkholderia, Pseudomonas genomes do not contain the genes to directly metabolize salicylate, however inoculation of Pseudomonas GM41 can up-regulate salicylic acid synthesis and degradation in Populus (Timm et al., 2016). Taken together, our data suggest that predicting plant phenotypic response to bacterial inoculation, even in overly simplified systems using fully sequenced bacterial strains, is extremely difficult.

Contrary to our predictions, leaf physiology (Fig. 3), plant height (Table S1), root:shoot (Table S1), specific leaf area (Table S1), specific root length (Table S1), and total plant dry mass (Fig. 2) were not influenced by bacterial inoculation. It is possible that multiple, overlapping plant signaling and gene expression effects induced by bacterial endophyte inoculation may mask a hosts’ phenotype response. For example, endophytes simultaneously up- and down-regulate numerous genes and metabolites in plant host (see Verhagen et al., 2004; Wang et al., 2005; Walker et al., 2011; Weston et al., 2012; Timm et al., 2016). Thus, counteracting influences among different gene pathways may conceal plant responses to endophyte inoculation when measuring down-stream phenotype and functional traits (Bashan, Holguin & de-Bashan, 2004; Timm et al., 2016). Additionally, host physiological response to endophyte inoculation may vary with bacterial strain (Kandasamy et al., 2009; Weston et al., 2012; Timm et al., 2016), plant host (Smith & Goodman, 1999), plant ontogeny (Siddiqui & Shaukat, 2003), or plant stress (Dimkpa, Weinand & Asch, 2009; Yang, Kloepper & Ryu, 2009; Lau & Lennon, 2012). For example, root colonization by Pseudomonas can reduce chlorophyll content and net photosynthesis (Asat) in a variety of plant hosts (Zou et al., 2005; Weston et al., 2012). However, Pseudomonas colonization can also increase photosynthetic activity and chlorophyll content (Kandasamy et al., 2009; Timm et al., 2016). Thus, biotic and abiotic contexts may drive the phenotypic response of hosts to endophyte inoculation, however this idea requires further testing.

Our study focused on the response of plant functional traits to monoculture associations of common endosphere bacteria, however future studies should focus on plant phenotype response to diverse microbiome communities. With a few well-known exceptions (Tan & Tan, 1986; Harris, Pacovsky & Paul, 1985; Ma et al., 2003; Lau & Lennon, 2011; Lau & Lennon, 2012), bacterial community composition in roots has been ignored in studies exploring what drives natural variation in plant traits (Friesen et al., 2011; Friesen, 2013; Timm et al., 2016). We propose a multifaceted approach to investigate linkages among the plant microbiome and natural plant trait variation. First, incorporation of microbiome composition into studies that currently investigate host identity/genotype and environmental parameters may be important for finding patterns in natural trait variation–especially when conducted across a variety of environmental gradients. Second, once correlations between microbiome composition and plant traits are observed in the field, detailed work constructing communities in the lab and greenhouse would enable a mechanistic understanding of what is underlying the observed patterns. These studies could be especially fruitful when conducted across natural biotic and abiotic environmental gradients in the laboratory, greenhouse, and field settings (Classen et al., 2015).

Conclusions

Our study demonstrates that bacteria living in plant roots can influence plant morphological traits. Increasingly, ecologists are using plant functional traits to explore how changing environments alter plant function (Wright et al., 2004; Reich, 2014). Plant traits, such as specific leaf area and specific root length, are often significantly correlated with important plant functions such as carbon fixation and nutrient uptake (Díaz & Cabido, 2001). Researchers are using correlations between plant traits and function to extrapolate how plants and ecosystems will respond to global change (Reich et al., 1999; Wright et al., 2004; Reich, 2014). While interactions between plant genotype and environment undoubtedly influence plant phenotypic plasticity (Bradshaw, 1965; Schlichting, 1986; Sultan, 2000; Des Marais, Hernandez & Juenger, 2013), phenotype is also heavily influenced by biotic factors, like microbiome bacterial endophytes (Lau & Lennon, 2011; Lau & Lennon, 2012; Wagner et al., 2014; Hacquard & Schadt, 2015). Given that plant-microbial studies, including ours, have observed strong linkages between microbiome and plant phenotype (reviewed in Friesen et al., 2011; Friesen, 2013) interactions among global change drivers, plant genotypes, and plant microbiomes, should be considered in trait-based approaches to ecological questions (Classen et al., 2015).

Supplemental Information

Supplemental Information 1 Linear mixed models results for the response of plant trait expression in response to microbial additions.

Physiology traits of Populus trichocarpa that were not inoculated with bacteria (no microbe control) (n = 8) (white) or were inoculated with Pseudomonas GM30 (n = 7), Pseudomonas GM41 (n = 8), or Burkholderia BT03 (n = 7). Across all treatments there were no significant differences in: a) Plant chlorophyll content (SPAD), b) ΦCO2 (expressed as the slope of carboxylase activity across different light levels), and c) carboxylase activity under maximum light level and CO2 concentration (Amax).

Click here for additional data file.

Supplemental Information 2 Morphological trait dataset.

Click here for additional data file.

Supplemental Information 3 Physiolology measurements (Carboxylase activity, Amax, and Asat).

Click here for additional data file.

Supplemental Information 4 Microbial colonization experiment in which we inoculated Populus trichocarpa seedlings with Pseudomonas GM30, Pseudomonas GM41, Burkholderia BT03, and compared colony forming units from microbial treatments and a non-inoculated control plant.

Click here for additional data file.

Supplemental Information 5 Data Analysis R code.

This R script includes morphologic traits, physiologic traits, and microbial colonization experiment data.

Click here for additional data file.

We thank Lee Gunter, Tse-Yuan Lu, Kelsey Carter, Jesse Labbe, and W. Nathan Cude for their assistance in data collection and plant propagation. The Classen Ecosystem Ecology group and three reviewers provided valuable input on earlier reviews of this manuscript.

Additional Information and Declarations

Competing Interests

Author Contributions

Data Deposition

The authors declare that they have no competing interests.

Jeremiah A. Henning conceived and designed the experiments, performed the experiments, analyzed the data, contributed reagents/materials/analysis tools, wrote the paper, prepared figures and/or tables, reviewed drafts of the paper.

David J. Weston conceived and designed the experiments, contributed reagents/materials/analysis tools, reviewed drafts of the paper.

Dale A. Pelletier conceived and designed the experiments, contributed reagents/materials/analysis tools, reviewed drafts of the paper.

Collin M. Timm performed the experiments, reviewed drafts of the paper.

Sara S. Jawdy performed the experiments, contributed reagents/materials/analysis tools, reviewed drafts of the paper.

Aimée T. Classen conceived and designed the experiments, analyzed the data, contributed reagents/materials/analysis tools, wrote the paper, reviewed drafts of the paper.

The following information was supplied regarding data availability:

The raw data has been supplied as Supplemental Dataset Files.

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
