# Peer review of "Root bacterial endophytes alter plant phenotype, but not physiology"

_PeerJ, doi:10.7717/peerj.2606_

## Round 0.1 · original submission · Minor Revisions

The two reviewers consider this a rigorous study that is of interest to the field, but have suggested a number of changes, mostly minor, all of which need to be addressed in a revised manuscript. In particular, the questions regarding the experimental design need to be addressed and the Discussion section needs to be shortened.

There is also a major issue raised by reviewer1, who asks that the experiments be repeated using a wider range of bacteria in a soil medium rather than agar. I request that you either extend the experiment as suggested and provide the new data, or provide a convincing argument why that is not necessary and defending the validity of the results.

Reviewer 1 ·

Basic reporting

: Henning and co-workers have presented a manuscript that is written in professional English in a style that is easy to follow. There are very few typographical errors (lines 59, 209, 291, 300, 359). The introduction laid out the background to the work and attempted to show a lacunae in the role of bacterial endophytes and changes in expression of their genes that has an influence on the phenotype of the plant. The figures generally conform to the stated. The figures are relevant but could be presented in a format that is easier to compare.

Experimental design

. It is stated how research fills an identified knowledge gap. Rigorous investigation performed to a high technical & ethical standard. Methods described with sufficient detail & information to replicate.
In this study Henning and co-workers used three endophytic bacteria to determine whether they were able to alter the host plant trait plasticity and physiology. These 3 bacteria (2 Pseudomonas species and 1 Burkholderia were isolated from the host plant, but it is not clear whether they were selected for this study because they were the only strains available or for some other reason. The authors claim to have chosen these strains as they had different in predicted carbon-metabolism breadth (this definition was not explained), plant hormone production (which hormones?) secondary metabolite production (this is expected as very few strains produce a similar suite of compounds) and predicted strain function (this can mean anything). Therefore while the questions asked are relevant, there is not enough diversity amongst the strains used to achieve a meaningful outcome. The methods are easy to replicate, but the photograph of the microcosm indicates that space is rather restrictive and may influence the plant plasticity in a different manner.
I also have an issue with the use of an agar type medium (Murashige and Skoog) for these studies rather than soil. The method of application of the bacteria has not been stated clearly.

Validity of the findings

The data obtained is robust has sufficient replicates and is statistically sound. However, because of the manner in which the experiments are set up (as described in EXPERIMENTAL DESIGN above) it is not surprising that the data obtained shows such a limited response. Is the elongation of a root caused by the increased level of plant hormones a change in phenotype. Technically it is. However, it is a common phenomenon that occurs with many PGPR strains. A change in root architecture is something quite different, but was not observed here. The increase in leaf surface could be considered to be a real change in plasticity. The 5-page Discussion is very lengthy, due to a lot of speculation, and could be shortened considerably. The authors need only to discuss the limited results they obtained.

Additional comments

There is merit in the approach taken but with the current experimental set up and strains used it is understandable that the results show a very limited range of response by the plant, I would like to see a wider range of bacteria applied in a soil system in order to obtain a lot more data on which to base your conclusions.

Reviewer 2 ·

Basic reporting

-The Introduction provides very little background regarding the term “plasticity”, which is one of the main foci of the article, and this led to a lot of confusion for me throughout the article. For example, in L38, the Friesen review paper actually reports that microbes modified 14/30 traits (i.e. resulted in plasticity of 14/30 traits). To state that microbes “modify the plasticity” of those traits is incorrect, because it suggests that plasticity was measured both in the presence and the absence of bacteria, and that plasticity is modified by the bacteria. A similar phrase (“how bacterial strains alter the plasticity of…”) is used in L62, but was not actually measured in this study. Plasticity in response to bacteria was measured in this study, but not how bacteria modify that plasticity. I think if the authors add their definition of plasticity, this would clear up some confusion.

-Similarly, the major take-home of the study, as referenced in the title of the article, is that bacterial strains “alter plant plasticity.” However, the study did not measure whether plasticity was altered by bacterial inoculation…it only measured whether plasticity did or did not occur in response to bacterial inoculation. Plasticity is generally defined as when a given genotype expresses different trait values under different conditions. For example, a plant that exhibits high SLA when inoculated with bacteria, but low SLA when grown in sterile culture, would be said to have exhibited plasticity in response to the bacteria. In this study, it is correct to say that bacterial inoculation resulted in trait plasticity (i.e. traits were different in the bacterial treatments than in the control). However, whether the bacteria altered trait plasticity is impossible to say from this study. Again, saying this would require assessing trait plasticity in the absence of bacteria (for example, assessing how traits plastically shift between low and high nitrogen conditions), and also in the presence of the bacteria (i.e. assessing how traits plastically shift between the low and high nitrogen conditions). Then it could be assessed whether plasticity in the absence of bacteria differed from plasticity in the presence of the bacteria. Perhaps the title could be reworded as “Root bacterial endophytes alter plant phenotype, but not physiology”. (since alterations in plasticity were not assessed here).

-It seems unusual to me that the authors have described the results of the figures in the Figure captions, rather than simply describing the contents of the figures in the figure captions. For example, in Figure2 caption, “GM30 inoculation increased new leaf production by 35%” would generally be written in the Results section, not in the Figure caption. But, that is likely up to the Journal and the Editor’s preference.

-The raw data and R code were included in the submission, but they should likely be given Supplementary Table numbers and referenced in the manuscript so the reader knows where to find them.

-The raw data tables should all include units in the column headers so that they are more interpretable.

Experimental design

-L114: what was the resolution (dpi) of the plant tissue scans?

-L137: For the bacterial isolates, the authors state that descriptions of the isolates can be found in a given set of references cited. However, the authors do not state how these isolates were obtained, or from what wild Populus species they were obtained from. If the Populus species were not Trichocarpa, it would be useful to add whether these bacterial strains actually associate with P. trichocarpa in natural environments (even though they clearly do in the controlled conditions of the present study).

-L169: How did the LI-6400 chamber settings differ between the Amax and the Asat measurements? In other words, if Asat was measured under light saturating conditions, what were the light settings for the Amax measurements? Wouldn’t saturating light also be required to measure the “maximum rate leaves were able to fix carbon (Amax)”?

-L209: Please state that “percent change in trait values” is how you are measuring trait plasticity, since trait plasticity is a major focus of the paper.

-What is the alpha value at which you are considering your statistical results to be significant?

Validity of the findings

-L266: This statement is incorrect. Not all strain types differed from the controls for leaf and/or root growth rate in Figure 2 (not all p-values are significant).

-L275: It should be noted that the p-values of 0.36 and 0.17 were not significant (assuming this study used a p-value threshold of 0.05), indicating that leaf number actually did not increase for two of the three bacterial strains. This type of statement (stating that the strains differ, even though p-values are non-significant) occurs several times throughout the Results section and should be clarified…for example, L283.

-L287: please add the t-values and p-values to support the statement that Pseudomonas GM30 increased leaf surface area and aboveground dry mass.

-L336: this should be changed to “…strains POTENTIALLY differed…”. This study did not actually measure the functions mentioned, but only assessed genomic differences related to those functions. Whether the genes corresponding to those functions were expressed or not under the conditions of the study is unknown.

-L368: this should be rephrased to “…the POTENTIALLY higher auxin…”, since auxin production was not measured in this study.

-L408: The only root trait that was generally influenced by microbes was root growth rate, so I think “root structure” should be rephrased as “root growth rate”.

Additional comments

Overall I thought this article did a nice job of reporting a well-executed, scientifically sound study. My main difficulties with the manuscript were with some of the language used to describe the background and results, as indicated in my comments above. I've also listed some more specific comments below and hope they are helpful.

-There is a typo in the title: should be “plasticity”

-L76: “stains” should be “strains”

-L216-L222: This paragraph describes results, and should be moved to the Results section.

-L243: This appears to be a sentence fragment. Should the sentence be rephrased as “…and MORE secondary metabolite biosynthesis elements…”? It also seems that it should be added that GM30 does NOT encode for phosphate solubilization and nitrate reduction elements, if I understood correctly what is being stated in L242.

-L253: “stains” should be “strains”.

-L257: “Pseudomonas” should be italicized throughout.

-L289: “out” should be “our”.

-L300: perhaps change “…leading to increases…” to “…linked with increased…”? It would be impossible to know from this study whether changes in plant structure were “leading” to changes in physiological traits. But, whether the two are linked can be reported.

-L322: please explain what is meant by “benefit the host plant”, given that total dry mass was not influenced by bacterial strains.

-L330: I think this sentence should be rephrased, since in this study, some belowground and aboveground traits were influenced by bacterial inoculation, while others were not. In other words, from what I understand of the results, there doesn’t seem to be an inherent difference in the way above vs belowground traits were affected in this study.

-L356: please define ‘generalist-specialist’ tradeoffs. I think this would help the reader to better understand its relevance to the study at hand.

-L359: “denitrification”

-L399 and 400: The way this sentence is currently phrased, it sounds as if the authors are presenting a result about different plant genotypes, but only one genotype was used in this study. Could you rephrase using something like “previous studies have shown that…”?

-L409: I was confused by the statement that root structure “has very little genetic or environmental control”. Could you please explain what controls root structure if not genetics or the environment? Was your statement simply distinguishing between the abiotic environment versus the biotic environment?

-L423: typo in “communities”.

-Figure 2 legend: 1138% is a typo.

-Supp. Table 1: “co-variety” should be “co-variate”.

---

## Round 0.2 · Minor Revisions

Thank you for revising the manuscript so comprehensively; it is much improved. I suggest one more minor change - while the figures look good with the raw data points added, I think you need to explain what you have done in the figure legends

---

## Round 0.3 · accepted · Accept

The figure legends amended as requested, thank you.